# Molecular Imaging and Nanotechnology—Emerging Tools in Diagnostics and Therapy

**DOI:** 10.3390/ijms23052658

**Published:** 2022-02-28

**Authors:** Marcin Woźniak, Agata Płoska, Anna Siekierzycka, Lawrence W. Dobrucki, Leszek Kalinowski, Iwona T. Dobrucki

**Affiliations:** 1Department of Medical Laboratory Diagnostics-Fahrenheit Biobank BBMRI.pl, Medical University of Gdansk, 80-210 Gdansk, Poland; marcin.wozniak@gumed.edu.pl (M.W.); agata.ploska@gumed.edu.pl (A.P.); anna.siekierzycka@gumed.edu.pl (A.S.); dobrucki@illinois.edu (L.W.D.); 2Beckman Institute for Advanced Science and Technology, University of Illinois at Urbana-Champaign, 405 N Mathews Ave, MC-251, Urbana, IL 61801, USA; 3Department of Neurobiology, Maj Institute of Pharmacology, Polish Academy of Sciences, 31-343 Krakow, Poland; 4Department of Bioengineering, University of Illinois at Urbana-Champaign, Urbana, IL 61801, USA; 5BioTechMed Centre, Department of Mechanics of Materials and Structures, University of Technology, 80-210 Gdansk, Poland

**Keywords:** molecular imaging, theranostics, personalized medicine, nanotechnology, nanoparticles, imaging modalities

## Abstract

Personalized medicine is emerging as a new goal in the diagnosis and treatment of diseases. This approach aims to establish differences between patients suffering from the same disease, which allows to choose the most effective treatment. Molecular imaging (MI) enables advanced insight into molecule interactions and disease pathology, improving the process of diagnosis and therapy and, for that reason, plays a crucial role in personalized medicine. Nanoparticles are widely used in MI techniques due to their size, high surface area to volume ratio, and multifunctional properties. After conjugation to specific ligands and drugs, nanoparticles can transport therapeutic compounds directly to their area of action and therefore may be used in theranostics—the simultaneous implementation of treatment and diagnostics. This review summarizes different MI techniques, including optical imaging, ultrasound imaging, magnetic resonance imaging, nuclear imaging, and computed tomography imaging with theranostics nanoparticles. Furthermore, it explores the potential use of constructs that enables multimodal imaging and track diseases in real time.

## 1. Introduction

In recent years, the vigorous growth of research techniques has led to a new scientific discipline area described as molecular imaging (MI). The MI approach provides a prospect of noninvasive real-time projection of many natural phenomena and processes in cell cultures and throughout the body at the cellular and molecular tiers [1,2,3,4]. Overall, MI demands implementing a particular apparatus either separately or together with a molecular feature that can depict individual tissues in the body and specific biochemical indicators. The acquired information can improve the knowledge about natural phenomena, determine the present pathologies, and deliver data on disease mechanisms. Moreover, MI has a tremendous perspective for the advancement of diagnostics, therapy, drug development, and deep insight into nanoscale processes, such as the interplay between proteins and enzymatic modifications. Together with the knowledge in genetics, genomics, and proteomics, MI is considered one of the pivotal foundations for the development of personalized medicine.

In the last decades, the scientific community has seen a growing interest in nanotechnology solutions and their subsequent applications, specifically in pharmacology, biomedicine, cosmetics, and the food industry [5,6]. Nanoparticles (NPs) are mainly developed based on carbon structures, metal and metal oxides, polymers, lipids, and semiconductors. Nanotechnology enables targeted delivery, improves stability in different environments and conditions of the gastrointestinal tract, solubility, and bioavailability [7,8]. These properties are essential for medicines used in pharmacology and active compounds in cosmetics and nutraceuticals in fortified food. Antimicrobial properties of nanomaterials based on silver and gold nanoparticles are used both in medicine (e.g., in silver-coated patches and bandages), personal care products and cosmetics, and in the food industry (e.g., in chicken farms or in food storage equipment) [9,10,11]. There is a significant number of clinical trials involving nanoparticles; according to the ClinicalTrials.gov database, over 500 clinical trials, including nanoparticles, have been registered up to today [12]. Some of the NPs have already been approved by the Food and Drug Administration (FDA) and/or European Medicines Agency (EMA) for use in humans, among them an improved pharmaceutical form of numerous anticancer drugs and antibodies, iron derivatives, bone substitutes, and recently, vaccines against COVID-19 [13,14,15,16,17].

However, the small sizes of nanoparticles and the ability of cell membrane penetration may cause some health issues. External dimensions in the lower range of the nanoscale, insolubility, specific morphological shape (e.g., needle shape and long rigid fibers), surface reactivity, the potential for radical formation, or other surface properties that can enhance cellular uptake, or allergenicity may cause toxic effects, including cell membrane damage, oxidative stress, inflammation, and even genotoxicity [13,16,18,19].

There are several guidelines for nanomaterial handling and databases that help to determine the properties and possible toxic effects of some nanomaterials, but the management of toxicity and safe handling of nanoparticles are still debated and should be well-concerning, especially before their application in oral and parenteral preparations [7,9,18,19,20,21].

In the context of molecular imaging, nanotechnology and nanomaterials, and in particular, nanoparticles, have great potential and constitute a new set of diagnostic tools. Due to their varied size, shape, composition, and exceptional surface properties and reactivity, nanoparticles are considered the most modifiable imaging agents. However, no well-established size determines the sizes of nanoparticles. Scientists use the term to describe structures ranging from 10 to 200 nm but not larger than 1000 nm [22]. Nanoparticles are larger than many proteins and small molecules but still smaller than cells [23]. Due to their relatively large sizes, they are absorbed by the endoplasmic reticulum. This phenomenon can be partially counteracted by covering the surfaces of nanoparticles with polymers. Due to the high surface area to volume ratio, it is possible to transfer target substances (ligands), signaling elements (fluorochromes and isotopes), and therapeutic agents utilizing nanoparticles. This means that developing a molecular probe in a nanoparticle with a particular binding site generates a strong signal, thus enabling precise detection of the cell’s specific structure. This anastomosis’s multivalent nature is directly related to avidity and is especially important when the problem is the imaging technique’s sensitivity (e.g., magnetic resonance imaging; MRI).

Moreover, many signaling elements conjugated to the single nanoparticle increase the likelihood of conjugating the construct at a specific binding site. Additionally, the ability to transport therapeutic compounds directly to their area of action may presumably reduce many undesirable effects of current therapies. Thus, by combining modern molecular biology methods, advances in chemical synthesis, imaging techniques, and the use of nanoparticles’ multifunctional properties, researchers are contributing to improving diagnostics and therapy, leading to personalized medicine development.

Personalized medicine is a concept based on understanding the differences between patients suffering from the same disease and understanding the complexity of diseases simultaneously. Thanks to this knowledge, it is possible to select appropriate therapies for specific groups of patients. Personalized medicine makes it possible to predict whether a particular therapy will be sufficient for a given patient. The principle of personalized medicine or “the right medicine for the right patient at the right time” has been practiced almost since the dawn of time. Already, Hippocrates (5th century BCE), based on Empedocles’ theory, recommended therapy depending on the disturbances of the body’s four basic moods. Historically, the first example of a personalized approach to therapy is transfusion medicine. It is worth emphasizing that the authors of the theory of inheritance of blood groups, which became the basis for the proper selection of blood donors and recipients, were Polish researchers—Anna and Ludwik Hirszfeld.

Historically, a scientist who can be considered a pioneer of personalized medicine is the American researcher of the renin–angiotensin system—John Laragh [24]. He showed that the response to selected antihypertensive drugs depends on the baseline activity of the renin–angiotensin system. The concept of Prof. Laragh has recently been confirmed in a study conducted in patients with resistant hypertension (PATHWAY-2), in which the antihypertensive efficacy of drugs acting through the renin–angiotensin system depended on the baseline plasma renin activity [25]. Personalized medicine also means close interactions between diagnosis and therapy, because these are precise diagnostic tools that help establish differences between patients suffering from the same disease and then adjust the treatment to specific groups of patients. An expression of this is the creation of a new term—theranostics—a combination of the words therapy and diagnostics. The objective of the theranostics approach is the simultaneous implementation of treatment and diagnostics.

This review will emphasize different imaging modalities with theranostics nanoparticle properties—both intrinsic and ones that require the inclusion of a drug into the designed nanoparticle (Table 1). Furthermore, we give special consideration and accentuate those constructs that provide information and track the disease in real time.

## 2. Literature Search Methodology

Three search engines (PubMed, Scopus, and Web of Science) were utilized to prepare the following manuscript with selected keywords: molecular imaging, nanotechnology, theranostics, personalized medicine, nanoparticles, and imaging modalities. The literature investigation scrutinized papers published only in English between 2001 and 2022. Additionally, three essential papers were included in the final literature review (1972, 1993, and 1994, respectively).

The overall amount of ~300 papers was preselected during the initial literature browsing. Six people divided into teams of two evaluated ~150 research articles that differed by the subject of interest, e.g., optical imaging, ultrasound, and nuclear imaging. Fleiss’ Kappa analysis was employed to evaluate the compliance between two evaluators (0 demonstrates no compliance, and 1 shows excellent compliance). A Kappa factor of >0.67 was applied (good compliance) as a threshold to incorporate a paper. The definitive version of this manuscript includes 68 research papers.

## 3. Optical Imaging

Presently, optical imaging is a novel and rapidly developing imaging technique that allows for noninvasive evaluation of the person’s organism with a cellular resolution. Fluorescence and bioluminescent imaging equipment are relatively cheap and easy to install. The system’s most important elements include the charge-coupled device (CCD) matrix and optical filters with a light source. These techniques allow achieving a perfect detection level—a range from picomolar to femtomolar concentrations. Additionally, these methods are considered relatively safe, because the radiation energy is relatively low compared to the gamma radiation emitted in positron emission tomography (PET) and single-photon emission tomography (SPECT). However, low-energy radiation means that the penetration depth is limited to only a few centimeters. For this reason, it is practically impossible to analyze structures in large animals and humans without the use of an endoscope, allowing you to get as close to the organs as possible. The use of a light source in the form of a laser with higher energy results in damaging the analyzed tissues or systems by high temperature [49]. The obstruction of low radiation penetration is not a problem in mice and small animals (not larger than a rabbit) due to their small sizes. As a result, many fluorescent and bioluminescent imaging markers can potentially visualize internal organs in small animals, making optical imaging techniques suitable for preclinical studies. The near-infrared-driven probes (700–900 nm) have been proposed for in vivo theranostic imaging to overcome these boundaries.

Hapuarachchige et al. established a pretargeting strategy merged with an image guidance technique to prevent the possible challenge with antibody–drug conjugates and their inseparable high toxicity [26,27]. In this paper, BrCa BT-474 cancer cells with HER2 overexpression were prestained with a functionalized trastuzumab monoclonal antibody and drug delivery unit. Both elements are conjugated using the biorthogonal click chemistry approach and incorporated as nanoclusters. Researchers have shown that the described strategy characterizes better therapeutic effectiveness compared to the treatment with a drug delivery element only. When combined with molecular imaging modalities, this therapeutic platform can become a powerful theranostics strategy that will ultimately direct cancer schematic evolution into accuracy, distinctiveness, and safety. In another study, scientists developed self-verifying and self-tailoring programmed theranostics compounds for cancer applications [28]. These novel incitement-receptive nanoparticles also contain an anticancer drug (camptothecin), targeted element (folate), and a caspase-3 triggerable fluorescent peptide (dabcyl-KFFFDEVDK-FAM). To turn off the fluorescence, researchers employed the resonance energy transfer (FRET) effect. On the other hand, the fluorescence signal is turned on following the reaction with caspase-3, which can track the apoptosis phenomenon after delivering the theranostics nanoparticle. In murine HeLa xenografts injected with the developed construct, the authors revealed the power of therapeutic tracking and provided semiquantitative data up to 48 h. Additionally, tumor volume growth inhibition was observed up to 15 days, indicating that the developed single-nanoparticle-driven platform is feasible for the therapeutic self-reporting visualization of cancer eradication.

As previously stated, researchers have developed numerous near-infrared (NIR) fluorescence probes for tracking physiological and pathological processes both in vitro and in vivo [50,51,52]. Using NIR fluorescence provides more excellent spatial resolution and sensitivity with a simultaneously reduced autofluorescence [53,54,55,56]. Zhu et al. depicted a NIR prodrug DCM-S-CPT and its nanoparticles packed with polyethylene glycol-polylactic acid (PEG-PLA) as a potent cancer therapy [30]. In this concept, a dicyanomethylene-4*H*-pyran (DCM) derivative serves as NIR dye and has been functionalized through a disulfide linker with camptothecin (CPT), an anticancer drug. During in vitro experiments, the authors proved that a high GSH concentration inside tumor cells generates a disulfide linker segmentation. This cleavage induces CPT therapeutic release and considerable fluorescence initiation simultaneously. Additionally, the developed prodrug has been effectively employed for the in vivo monitoring of drug release and anticancer therapeutic performance using NIR fluorescence. PEG-PLA nanoparticles incorporating DCM-S-CPT demonstrate higher anticancer properties than free CPT and have more prolonged blood clearance. Thus, DCM-S-CPT becomes an encouraging prodrug candidate offering meaningful progress into the better comprehension and exploration of theranostics drug delivery strategies.

Researchers have paid particular attention to small colloidal semiconductor nanocrystals, quantum dots (QDs), in recent years. Due to nanometer sizes, longer duration of fluorescence, better photostability, and a narrow emission spectrum, QDs are much more stable and precise fluorescent markers than those used so far in medical diagnostics: organic dyes [57]. Another advantage of QDs is that they can be used for imaging several molecular targets simultaneously, which is vital in cancer diagnostics in which numerous genes and proteins are involved [58]. They can also be utilized during in vitro diagnostic tests, where they enable the detection of many tumor biomarkers simultaneously, e.g., in serum [59]. QDs become the basis for the creation of multifunctional nanoparticles. It is possible connecting QDs with specific antibodies that recognize the antigen on neoplastic cells. The surfaces of quantum dots enable attaching molecules with therapeutic activity, providing, at the same time, drug transport and in vivo imaging [60].

Wang et al. developed pH-sensitive CdSe/ZnS-QDs conjugated with an anticancer drug, doxorubicin (DOX) [61]. Nanoparticles release the drug when the pH inside the cell milieu decreases. The basis of this strategy is the knowledge that cancer cells are characterized by lower pH compared to healthy cells. The proposed nanoplatform showed significant DOX discharge (65%) following incubation in a buffer at pH 5.0, which mitigates the cancer tissue. Although the experiment was conducted at pH 7.4 (healthy tissue), the Dox release was notably diminished (10%). The major limitation of the described study was the absence of in vitro and/or in vivo evaluations of the developed construct.

Paul et al. utilized silicon QDs to create a photosensitive nanocarrier that employed *o*-nitrobenzyl as a light-driven activation system for regulated anticancer drug (chlorambucil) distribution [29]. The idle condition was granted by trapping chlorambucil and suppressing QDs fluorescence signal through *o*-nitrobenzyl. As a result of QDs molecules excitation, the caged drug was released, and the fluorescence signal was detectable, enabling real-time tracking of the drug administration. The authors demonstrated in HeLa cells that the proposed system was stable for 30 min.

## 4. Ultrasound Imaging

Ultrasound is considered one of the most popular diagnostic techniques. In recent decades, due to the rapid development of ultrasound equipment, the ultrasound wave itself, its physical properties, and propagation laws are currently the main limitation in the quality of ultrasound imaging. With frequencies in the 1–10-MHz range and an ability to focus the beam in a small area, short acoustic waves have found many applications in biomedicine [62]. Most often, ultrasound waves are used for diagnostic imaging: ultrasound and ultrasound tomography. Ultrasound is also used for therapeutic purposes, e.g., thermal therapies (hyperthermia and thermoablation). The efficiency of ultrasound therapies can be improved by using special sound-active materials, e.g., nanoparticles [63]. Additionally, the contrast of ultrasound imaging can be improved with nanomaterials. The same nanomaterials can simultaneously influence ultrasound therapies and imaging, making them suitable for ultrasound therapy.

Researchers from Taiwan showed DOX and SuperParamagnetic Iron Oxide (SPIO) Nanoparticles caged in microbubbles (Mbs)—DOX-SPIO-MBs—for brain tumor treatment [34]. The proposed approach uses focused ultrasound to loosen the brain–blood barrier and exact drug administration. DOX-SPIO-MBs served as a dual-modality MRI and ultrasound contrast agent and provided magnetic targeting to improve DOX delivery. The authors demonstrated in a rat glioma model enhanced DOX-SPIO-MB accumulation in tumor tissue (22.4%) and durably yielded a meaningful superparamagnetic/photo-acoustic imaging contrast agent.

Scientists from Stanford University created a theranostics silica-made nanoparticle filled with insulin-like growth factor [35]. They suggested that a developed nanocarrier can boost cardiac stem cell growth for heart disease therapy. The impedance discrepancy between tissue and the silica nanoconstruct allows strengthened cell monitoring and molecular imaging in vivo. Additionally, the nanoparticles were equipped with gadolinium, which utilizes synthesized structures in other high-resolution imaging modalities. Human bone marrow mesenchymal stem cells were loaded with silica nanoparticles and embedded into nude mice’s left ventricle wall. Ultrasound images revealed intensified scatter in contrast to vehicle-only groups. Researchers determined that cell numbers down to 10,000 and 100,000 are detectable using ultrasound or MRI, respectively. Significantly, 1-week post-stem cell implementation, viability, and survival were improved up to 40%; the nanoparticles deteriorated in cells within 1 month after their initial administration.

Gao et al. synthesized oxygen-producing MnO_2_ nanoparticles, which can be tracked by ultrasound regarding photodynamic therapy [36]. Photodynamic therapy assumes that a photosensitizer generates reactive oxygen species (mainly H_2_O_2_) to initiate cell degradation. Commonly, tumor cells are characterized by an enhanced production of H_2_O_2_; thus, the authors synthesized oxygen-producing nanoparticles targeted at cancer, allowing to monitor the adequate oxygen production in diseased tissue without beginning photodynamic therapy. The developed construct comprised hyaluronic acid attached to MnO_2_ nanoparticles and conjugated with indocyanine green (ICG) that served as a NIR—excited photosensitizer. Hyaluronic acid enhanced tumor targeting by binding with CD44 antigen-expressing cancer cells and administering MnO_2_ molecules upon hyaluronidase deterioration. Experiments performed on the squamous cell carcinoma cell line (SSC7) demonstrated oxygen production with uninterrupted NIR excitement for up to 10 h and increased the cytotoxicity in the evaluated cell line. In murine SCC7 xenografts, 6 h after nanoparticle injection, the NIR light source was implemented. As a result of the conducted experiments, tumor ablation was observed, suggesting the potential use of developed nanoparticles for in vivo tracking and image-directed therapy.

Lee et al. proposed a novel theranostics strategy engaging a nanodroplet platform that transitions into microbubbles, allowing siRNA therapy [37]. The nanodroplets conjugated with siRNA have a size of ~257 nm. After contact with ultrasound waves, the initial droplets encounter a transition to form gas microbubbles of 3822 nm. In human primary breast and lung cancer cell lines, researchers noticed a four-fold decreasing in cell viability, which confirms the effective administration of siRNA and subsequent gene silencing.

## 5. Magnetic Resonance Imaging

From a clinical perspective, MRI is one of the most important noninvasive diagnostic tools for disease monitoring [64,65]. MRI is characterized by excellent spatial resolution, but it is less sensitive than fluorescent imaging. In recent years, there has been visible progress in the development of nanoparticle systems that allows for the improvement of imaging and diagnostics by MRI [64,65,66]. The use of nanoparticles in MRI achieves greater contrast, which allows for the better differentiation of pathologically altered tissues from healthy tissues. Nanotechnology, in this aspect, is represented by inorganic nanoparticles of iron, gold, cobalt oxide, or incorporated nanoparticles of gadolinium.

Recently, the Xin Zhou group developed a multipurpose liposome with incorporating gadolinium-DOTA (MRI contrast agent) and functionalized with α_v_β_3_ integrin (targeted peptide) and paclitaxel (anticancer drug) [39]. Synthesized nanoparticles conquered paclitaxel insolubility, enhanced drug transport efficacy to the tumor, and reduced adverse symptoms. In vitro studies conducted in the A549 cancer cell line showed significantly improved cytotoxicity effect when incubated with the liposome-based nanoparticles. MRI revealed 16-times intensified T1 relaxivity in a cancer cell culture treated with functionalized liposomes compared to the vehicle. During in vivo experiments, tumor-bearing mice were injected with previously synthesized multifunctional nanoparticles and visualized using MRI. Researchers observed the excellent inhibition of tumor growth after nanoparticle administration.

Scientists from China and Singapore invented a novel MRI probe for drug release tracking. Their strategy was based on a combination of a photothermal core (Au-Cu9S5) and a paramagnetic ion/drug charging silica cover with a thermal-sensitive valve [40]. Encapsulated in silica shells, due to the NIR-II photothermal effect initiated in the core, they were released from the nanoparticle. Performed in vitro and in vivo studies indicated that phase change material facilitates synergistic chemotherapy and mild hyperthermia by avoiding early drug release and maintaining the regional temperature beneath 45 °C. Moreover, conducted MRI imaging studies have proven a positive signal correlation between paramagnetic ions and a released drug, providing noninvasive drug release tracking in real time.

In another research, Gue et al. presented a novel multifunctional nanoenzyme-driven theranostics molecule—PPy@MnO_2_-BSA(Ce6), which was designed to allow MRI-controlled photothermal and photodynamic therapy (PTT and PDT, respectively) in cancer [31]. The nanoparticles were prepared by polymerizing pyrrole and subsequent oxidization with high-valent manganese salts (potassium permanganate; KMnO_4_). Next, the complex was stabilized with bovine serum albumin (BSA). However, the synthesis’ major point was functionalization with chlorine 6 (Ce6), which provided the whole nanoparticle with unique and essential functions (a production of oxygen and reactive oxygen species and conversion photons to heat). PPy@MnO2-BSA(Ce6) could become an intelligent nanoparticle to track and direct the accurate cancer milieu with the listed features.

## 6. Computed Tomography

Computed tomography (CT) is a recognized and broadly used technique that allows for tissue spatial imaging, providing detailed anatomical visualization. However, it is worth noting that CT scan is characterized by a lower accuracy of soft tissue imaging, where a MRI has a significant advantage. In CT scanners, the X-ray tube emitting X-rays moves around the examined object, and changes in the intensity of radiation after passing through the scanning objects are recorded by detectors located around the perimeter of the device. Then, the obtained measurement values are processed electronically, and image reconstruction is obtained. Several imaging probes found the application in clinical practice, such as gold, iodine, bismuth salts, lanthanide, and iron oxide. Many of the listed materials attenuating X-rays are encapsulated in nanoparticle platforms, e.g., lipoproteins, liposomes, or polymeric compounds [67,68,69,70,71,72,73].

Korean researchers fabricated gold nanoparticles (GNP) designed for the simultaneous imaging and therapy of prostate cancer [41]. RNA aptamer with prostate-specific membrane antigen (PSMA) was conjugated to obtain the GNP-based nanostructure’s targeting properties. An in vitro evaluation showed that PSMA-targeted GNP generated significantly higher CT image intensity in LNCaP cells (PSMA high expression) than a PC3 (PSMA low expression) cell culture. Furthermore, after DOX encapsulation within GNP, LNCAP cells revealed a higher sensitivity than PC3 cells to DOX, consistent with the previous findings. However, the study’s most significant limitation is the lack of performed experiments in cancer animal models.

Yang et al. synthesized 1,2-dilauroyl-sn-glycero-3-phosphocholine (DLPC) functionalized with bismuth-based nanoparticles (Bi@DLPC) feasible for photothermal therapy (PTT) induced by a NIR light source simultaneously monitored under photoacoustic (PA) and CT imaging [38]. Bi@DLPC is characterized by properties beneficial for PTT and CT imaging due to the excellent photothermal conversion capability and bismuth elements. Conducted in vivo and in vitro experiments indicated Bi@DLPC efficacious uptake and accumulation in MDA-MB-231 cells. Performed PTT resulted in inhibition of the tumor growth without injuring the surrounding tissues and organs, giving clinical application potential.

## 7. Nuclear Imaging—PET and SPECT

Molecular imaging of living organisms has strong connections with nuclear medicine. Since the beginning of the field, its goal has been to improve the noninvasive diagnosis and treatment of patients using imaging equipment and radioisotopes both unconjugated and conjugated to molecules specific to chosen cellular structures. Combining PET and SPECT techniques with new isotope-labeled molecular probes has ushered nuclear medicine into a new era directly related to imaging. In the last decade, the analysis of biological phenomena and processes using imaging have become more common and are now not limited to radionuclide injections. In the rising age of enhanced individualization of treatments, nanostructures can become a remarkably beneficial therapy instrument and monitoring disease.

Nuclear imaging requires two particular radioisotopes: for SPECT—radionuclides that emit gamma radiation (γ-), and PET—elements releasing positrons. Globally, for the past decades, SPECT imaging has been considered a backbone of clinical nuclear medicine. It broadly utilizes technetium-99m (^99m^Tc) due to its advantageous physical features (6-h half-life and energy emission at 140 keV) that suites scintillation crystal-based gamma cameras. A worthwhile mention is that ^99m^Tc could be produced “on site” using a ^99^Mo/^99m^Tc isotope generator. To date, several liposomal nanoparticles have been labeled with ^99m^Tc radionuclides to detect sentinel lymph nodes and cancer imaging purposes [48,74,75,76,77,78,79]. One of the widely used platforms for theranostics applications is branched synthetic nanopolymers—dendrimers, conjugated with radioisotopes capable of emitting both β and γ rays concurrently (^131^I, ^177^Lu, and ^188^Re) [47,80,81,82]. Zhao et al. developed a multifunctional dendrimer-based nanomolecule labeled with ^131^I radioisotope for targeted SPECT imaging. The nanoparticle was functionalized with a specific compound, chlorotoxin, to enable in vivo matrix metalloproteinase-2-targeting glioma-bearing murine animal models. Moreover, dendrimers loaded with iodine serve as remarkable computer tomography (CT) contrast agents that accommodate multimodal imaging (e.g., SPECT and CT).

Goins et al. designed a liposome-based nanostructure with embedded radionuclide—rhenium-186 (^186^Re). ^186^Re is considered an excellent theranostics isotope that exhibits a favorable half-life of 89.3 h, perfect for the diagnostic, treatment, and monitoring of the disease. Additionally, it is characterized by gamma energy emission at 137 keV, which suits optimally for SPECT imaging. By dint of a prolonged half-life, imaging can be conducted a couple of days after nanoparticle injection. The radiolabeled nanoliposome’s beta radiation path length covers a distance of 2 mm, meaning the construct needs to be delivered only to the cell’s region without being endocytosed.

PET imaging is a method of molecular tissue imaging widely utilized in the diagnosis of various abnormalities providing quantitative data. The basis of this technique is the phenomenon of positron–electron annihilation, resulting in the formation of two high-energy photons (511 keV) emitted in opposite directions (180°). Positrons are derived from the decay of a radioactive isotope that is a component of an administered radiopharmaceutical. The source of electrons is the examined subject’s tissues and body fluids.

Dobrucki et al. developed marker ^64^Cu-NOTA-PEG_4_-cRGD_2_ that binds to the α_V_β_3_ integrin receptor, allowing to visualize the active process of angiogenesis in vivo with the PET technique. This marker is based on a structure of dimeric cyclic peptide containing the Arg-Gly-Asp (RGD) sequence labeled with NOTA (1,4,7-triazacyclononane-N,N’,N”-triacetic acid) that is capable of binding radionuclides like ^64^Cu [42]. ^64^Cu-NOTA-PEG_4_-cRGD_2_ was effective for imaging ischemia-induced angiogenesis in animal models of myocardial infarction or hindlimb ischemia (HLI) and increased the angiogenesis in xenografts of tumor-bearing mice [42,43,44].

The second marker for PET imaging synthesized by the combined Dobrucki and Kalinowski research group was MMIA (multimodal imaging agent) for imaging of the receptor for advanced glycation end products (RAGE). This nanoparticle was based on a fourth-generation polyamidoamine-dendrimer (G4 PAMAM) structure that was also conjugated to NOTA binding ^64^Cu for PET imaging, with fluorophore (rhodamine) allowing for optical detection and with the ligand specific for RAGE—carboxymethyl-lysine (CML)-modified human serum albumin (HSA). MMIA was used to visualize the ischemic region in mice after HLI (Figure 1) [32,33].

Dr. Cai’s research team developed a mesoporous silica nanoparticle-based platform functionalized with an antibody specific to CD-105, labeled with ^64^Cu for in vivo PET imaging and tracking, and containing a therapeutic agent—doxorubicin [45]. PET imaging performed in 4T1 murine breast cancer xenografts showed prompt and constant deposition nanoconstructs in cancer tissue. The authors suggested that the mechanism responsible for this is the enhanced permeability and retention in cancer cells and specific uptake through binding to the CD105 antigen overexpressed in tumor vessels. Researchers have observed efficient doxorubicin delivery to tumor tissue, proving the usefulness of the chosen nanoplatforms in the theranostics approach. Furthermore, the presented approach could be modified to target other cell structures, tumor types, and to synthesize multimodal imaging probes [83,84].

Liu et al. synthesized a gold nanostar (GNS) agent comprising multimodal properties involving X-ray-computed tomography (CT), surface-enhanced Raman scattering (SERS) identification, two-photon luminescence (TPL) imaging, photothermal therapy (PTT), and also labeled with ^131^I [46]. The scientists evaluated the yield for in vitro and in vivo photothermal heating and ablation (respectively) of primary sarcomas in mice. The authors indicated that smaller GNS (30 nm vs. 60 nm) present a better tumor intake and profound tumor tissue infiltration. Moreover, the GNS-injected dose was directly proportional to the ratio of tumor uptake.

## 8. Conclusions and Future Perspective

The turn of the 20th and 21st centuries has undoubtedly brought enormous progress to biomedical sciences and, at the same time, in diagnosing and treating various abnormalities. Thanks to the combination of modern methods of molecular biology, advances in chemical synthesis, and the use of nanoparticles’ multifunctional properties, we are getting closer to personalized medicine implemented with theranostics’ help that provides precise and early diagnostics and therapy at the same time. To accomplish these goals, methods with at least a sensitivity of 10^−5^–10^−7^ are needed. Such sensitive technology can allow for regular monitoring of the patient’s treatment and early detection of the disease. Currently, the used methods are not sensitive enough to detect the early spread of the disease. Therefore, this is the cause of disease recurrence, despite the complete removal of even a small primary tumor and the use of postoperative chemotherapy. It has long been suspected that a primary tumor releases cancer cells into the bloodstream very early in the disease before it becomes clinically symptomatic. A susceptible and specific technology will allow the future assessment of the stage of cancer at the molecular and clinically asymptomatic levels, giving the oncologist information and time to plan an effective treatment. Thus, in the scientific world, work on multifunctional nanoparticles that can freely penetrate cells and cause a specific biological effect is carried out very intensively. Currently, we are witnessing tremendous progress in diagnosis and therapy. The concept of personalized medicine is coming true; however, there is still much to be done, and advances in our knowledge of knitted therapeutic and diagnostic probes are needed to Improve its potential for clinical purpose.

## Figures and Tables

**Figure 1 ijms-23-02658-f001:**
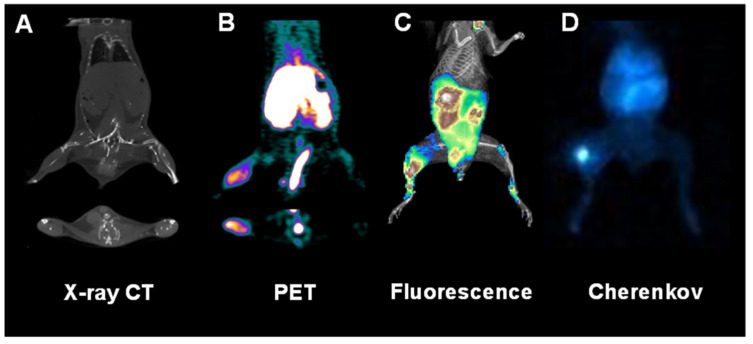
Multimodal multifunctional nanoparticles allow the in vivo imaging of anatomy, physiology, and molecular events at various spatiotemporal scales. Shown here are representative images of a mouse subjected to surgical ligation of the right femoral artery to induce hindlimb ischemia followed by the inflammatory response, which is assessed with a receptor for an advanced glycation end product (RAGE)-targeted nanoparticle-based multimodal agent labeled with both fluorophore (rhodamine) and radioisotope (^64^Cu). The anatomy was assessed with X-ray computed tomography (CT) imaging (**A**). In contrast, molecular proinflammatory events were quantitatively assessed in vivo with positron emission tomographic (PET) imaging (**B**), whole-body fluorescence (**C**), and Cherenkov luminescence (**D**).

**Table 1 ijms-23-02658-t001:** Summary of the depicted studies. Multimodal nanoparticles are listed multiple times.

Imaging Modality	Nanoparticle	Application	Therapeutic Component	Reference
Optical imaging	Albumin nanocarrier	Cancer	Trastuzumab	[26,27]
	Polymer nanoassembly	Cancer	Camptothecin	[28]
	Silicon QDs	Cancer	Chlorambucil	[29]
	PEG-PLA	Cancer	Camptothecin	[30]
	PPy@MnO_2_-BSA(Ce6)	Cancer	Photothermal and photodynamic therapy	[31]
	Bi@DLPC	Cancer	Photothermal therapy	[32]
	Dendrimer conjugated with CML, labeled with ^64^Cu and rhodamine	Peripheral arterial disease	------------	[32,33]
Ultrasound	SPIO trapped in MBs	Cancer	Coxorubicin	[34]
	Silica nanocarrier	Cardiac stem cell therapy	Insulin-like growth factor	[35]
	MnO_2_ functionalized with hyaluronic acid	Cancer	Photodynamic therapy employing indocyanine green as a photosensitizer	[36]
	Chitosan-deoxycholic acid, containing perfluoropentane and iron oxide	Cancer	siRNA	[37]
	Bi@DLPC	Cancer	Photothermal therapy	[38]
MRI	SPIO trapped in MBs	Cancer	Doxorubicin	[34]
	Silica nanocarrier	Cardiac stem cell therapy	Insulin-like growth factor	[35]
	Chitosan-deoxycholic acid, containing perfluoropentane and iron oxide	Cancer	siRNA	[37]
	Gadolinium loaded liposome	Cancer	Paclitaxel	[39]
	Mesoporous silica nanomaterial with embedded Au-Cu_9_S_5_	Cancer	Doxorubicin	[40]
	PPy@MnO_2_-BSA(Ce6)	Cancer	Photothermal and photodynamic therapy	[31]
CT	Gold nanoparticles conjugated with RNA aptamer with prostate-specific membrane antigen	Cancer	Doxorubicin	[41]
	Bi@DLPC	Cancer	Photothermal therapy	[38]
PET	RGD Peptide conjugated with NOTA, and labeled with ^64^Cu	Cancer, Myocardial infarction, and Peripheral arterial disease	------------	[42,43,44]
	Dendrimer conjugated with CML, labeled with ^64^Cu and rhodamine	Peripheral arterial disease	------------	[32,33]
	Mesoporous silica functionalized with an antibody specific to CD-105, and labeled with ^64^Cu	Cancer	Doxorubicin	[45]
	Gold nanostar	Cancer	Phototermal therapy	[46]
SPECT	Dendrimer functionalized with chlorotoxin and ^131^I	Cancer	Radiotherapy	[47]
	Liposome-based nanostructure with embedded ^186^Re	Cancer	Radiotherapy	[48]

## Data Availability

Not applicable.

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
