# Peer review of "Molecular Imaging and Nanotechnology—Emerging Tools in Diagnostics and Therapy"

_ijms, 2022, doi:10.3390/ijms23052658_

Round 1

Reviewer 1 Report

I suggest to change the type of paper from "Review" to "Perspective".

The authors should better mark the focus and the novelty character of this paper.

In Introduction the authors should better describe the application of nanotechnologies and nanomaterials in different fields, with particular regards to pharmaceutical, nutraceutical ones. Also safety aspects should be considered. References related to this matter should be added.

A section methodology should be added indicating the bibliographic research criteria.

A Table indicating the main and representative studies of Optical Imaging, Ultrasound Imaging, Magnetic Resonance Imaging, Computed Tomography, Nuclear Imaging – PET and SPECT, related to nanotechnologies, should be added. Comparative considerations between the use of different techniques are welcome.

Figures on the different techniques  are welcome.

Author Response

Dear Sir or Madame,

Thank you very much for the comments regarding our manuscript. In line with your suggestions we have made the appropriate amendments in the body of the manuscript. We believe all the comments addressed based on your feedback enhanced the quality of the review. We would like to thank you for your time and thoughtful suggestions.

We have addressed your comments on a point-by-point basis:

  1. I suggest to change the type of paper from "Review" to "Perspective".

Answer: Thank you for your comment. The request of changing the type of paper from "Review" to "Perspective" will be sent to the Editor of International Journal of Molecular Sciences Special Issue "Molecular Imaging in Nanomedical Research 2.0".

  1. The authors should better mark the focus and the novelty character of this paper.

Answer: Thank you for your comment. In recent years, the field of nanotechnologies, especially nanomedicine, has been widely explored in scientific journals. The novelty of our manuscript focuses on the multi-modality of nanoparticles used in molecular imaging and their’ potential for theranostics and personalized medicine, which we underline in sentence that was added at Abstract section (page 1, lines 29-30):

"Furthermore, it explores the potential use of constructs that enables multi-modal imaging and track the disease in real-time".

  1. In Introduction the authors should better describe the application of nanotechnologies and nanomaterials in different fields, with particular regards to pharmaceutical, nutraceutical ones. Also safety aspects should be considered. References related to this matter should be added.

Answer: Thank you for your comment. The extra paragraph was added regarding some applications in which nanoparticles are used and safety issues that should be considered during nanoparticle handling (page 2, lines 49 to 75).

"In the last decades, the scientific community has seen a growing interest in nano-technology solutions and their subsequent biomedicine applications specifically in pharmacology, biomedicine, cosmetics, and food industry[5,6]. Nanoparticles (NPs) are mainly developed based on carbon structures, metal and metal oxides, polymers, lipids and semiconductors. Nanotechnology enables targeted delivery, improves stability in different environment and conditions of the gastrointestinal tract, solubility and bioa-vailability[7,8]. These properties are essential for medicines used in pharmacology and active compounds in cosmetics and nutraceuticals in fortified food. Antimicrobial properties of nanomaterials based on silver and gold nanoparticles are used both in medicine (e.g., in silver-coated patches, bandages), personal care products and cosmetics, and in the food industry (e.g., in chicken farms or in food storage equipment)[9-11]. There is a significant number of clinical trials involving nanoparticles; according to the ClinicalTrials.gov database, over 500 clinical trials, including nanoparticles, have been registered up today [12]. Some of the NPs have already been approved by Food and Drug Administration (FDA) and/or European Medicines Agency (EMA) for use in hu-mans, among them improved pharmaceutical form of numerous anticancer drugs and antibodies, iron derivatives, bone substitutes, and recently vaccines against COVID-19[13-17].

However, the small size of nanoparticles and the ability for cell membrane penetration may cause some health issues. External dimensions in the lower range of the nanoscale, insolubility, specific morphological shape (e.g., needle shape, long rigid fibres), surface reactivity, the potential for radical formation, or other surface properties that can enhance cellular uptake, or allergenicity may cause toxic effects including cell membrane damage, oxidative stress, inflammation, and even genotoxicity[13,16,18,19].

There are several guidelines for nanomaterials handling and databases that help to determine properties and possible toxic effects of some nanomaterials, but the management of toxicity and safe handling of nanoparticles are still debated and should be well concerned, especially before their application in oral and parenteral preparations. [7,9,18-21]."

16 new references were included (page 13, lines from 487-521).

  1. A section methodology should be added indicating the bibliographic research criteria.

Answer: Thank you for your comments, we appreciate the feedback. We added methodology section explaining the literature search criteria that have been employed. (pages 5-6, lines 132-145):

"2. Literature search methodology

Three search engines (PubMed, Scopus, and Web of Science) were utilized to pre-pare the following manuscript with selected keywords: molecular imaging, nanotech-nology, theranostics, personalized medicine, nanoparticles, imaging modalities. The literature investigation scrutinized papers published only in English between 2001 to 2022. Also, three essential papers were included in the final literature review (1972, 1993 and 1994, respectively).

The overall of ~300 papers was pre-selected in the initial literature browsing. 6 people divided into teams of 2 evaluated ~150 research articles differed by the subject of interest, e.g., optical imaging, ultrasound, and nuclear imaging. Fleiss’ Kappa analysis was employed to evaluate the compliance between two evaluators (0 demonstrates no compliance and 1 shows excellent compliance). A Kappa factor of > 0.67 was applied (good compliance) as a threshold to incorporate a paper. The definitive version of this manuscript includes 68 research papers."

  1. A Table indicating the main and representative studies of Optical Imaging, Ultrasound Imaging, Magnetic Resonance Imaging, Computed Tomography, Nuclear Imaging – PET and SPECT, related to nanotechnologies, should be added. Comparative considerations between the use of different techniques are welcome. Figures on the different techniques are welcome.

Answer: Thank you for your comment. Table summarizing representative studies of imaging with different techniques discussed in our manuscript was added (pages 3-5, lines 128-130).

We decided not to add comparative considerations between the use of different molecular imaging techniques; some of the advantages and properties characterizing each technique have been already mentioned in sections regarding those techniques individually.

We decided not to add figures on the different techniques. Although we agree, that extra figures would add value to manuscript, we would rather attract the focus of potential reader to the beneficial use of multi-modal imaging agents.

Please see the attached revised version of our manuscript.

Reviewer 2 Report

The manuscript is a well-written narrative overview about the current and potential future application of MI with a theranostic approach, with a vision directed to implement personalized medicine

Only a minor issue:

Lines 74-89 (from "Personalized medicine... to ..plasma renin activity): I think that this paragraph does not add relevant information to the remaining manuscript, and this historical overview only leads away from the focus and the concept of the manuscript. It is really necessary?

Author Response

Sir or Madame,

Thank you for reviewing the manuscript title: “Molecular imaging and nanotechnology – emerging tools in diagnostics and therapy".

We would like to thank you for your time and thoughtful suggestion.

Thank you very much for the comment regarding our manuscript.

  1. Lines 74-89 (from "Personalized medicine... to ..plasma renin activity): I think that this paragraph does not add relevant information to the remaining manuscript, and this historical overview only leads away from the focus and the concept of the manuscript. It is really necessary?

Answer: Thank you for your comments, we appreciate the feedback. We are aware that paragraph in lines 74-89 does not add any relevant information to the issue of nanoparticles in molecular imaging. However, this brief historical overview gives the readers a broader perspective on personalized medicine. By including this, we want to emphasize that the concept of personalized medicine has been known over the centuries and is not strictly related to modern technologies that have been developed recently.

We would also like to note that some extra paragraphs and references were added to the manuscript due to another reviewer’s request. All introduced changes are listed below and marked up in the manuscript in “Track Changes” mode.

At Abstract section (page 1, lines 29-30), sentence that was added:

"Furthermore, it explores the potential use of constructs that enables multi-modal imaging and track the disease in real-time".

The extra paragraph was added regarding some applications in which nanoparticles are used and safety issues that should be considered during nanoparticle handling (page 2, lines 49 to 75):

"In the last decades, the scientific community has seen a growing interest in nano-technology solutions and their subsequent biomedicine applications specifically in pharmacology, biomedicine, cosmetics, and food industry[5,6]. Nanoparticles (NPs) are mainly developed based on carbon structures, metal and metal oxides, polymers, lipids and semiconductors. Nanotechnology enables targeted delivery, improves stability in different environment and conditions of the gastrointestinal tract, solubility and bioa-vailability[7,8]. These properties are essential for medicines used in pharmacology and active compounds in cosmetics and nutraceuticals in fortified food. Antimicrobial properties of nanomaterials based on silver and gold nanoparticles are used both in medicine (e.g., in silver-coated patches, bandages), personal care products and cosmetics, and in the food industry (e.g., in chicken farms or in food storage equipment)[9-11]. There is a significant number of clinical trials involving nanoparticles; according to the ClinicalTrials.gov database, over 500 clinical trials, including nanoparticles, have been registered up today [12]. Some of the NPs have already been approved by Food and Drug Administration (FDA) and/or European Medicines Agency (EMA) for use in hu-mans, among them improved pharmaceutical form of numerous anticancer drugs and antibodies, iron derivatives, bone substitutes, and recently vaccines against COVID-19[13-17].

However, the small size of nanoparticles and the ability for cell membrane penetration may cause some health issues. External dimensions in the lower range of the nanoscale, insolubility, specific morphological shape (e.g., needle shape, long rigid fibres), surface reactivity, the potential for radical formation, or other surface properties that can enhance cellular uptake, or allergenicity may cause toxic effects including cell membrane damage, oxidative stress, inflammation, and even genotoxicity[13,16,18,19].

There are several guidelines for nanomaterials handling and databases that help to determine properties and possible toxic effects of some nanomaterials, but the management of toxicity and safe handling of nanoparticles are still debated and should be well concerned, especially before their application in oral and parenteral preparations. [7,9,18-21]."

16 new references were included (page 13, lines from 487-521).

Literature search methodology section (pages 5-6, lines 132-145) and a table summarizing representative studies of imaging with different techniques discussed in our manuscript were added (pages 3-5, lines 128-130).

Please see the attached revised version of our manuscript.
